TECHNICAL RELEASE

# Efficient downsampling of genome alignments with Rasusa

Achmad Dimas Cahyaning Furqon[1], Leah W. Roberts[1] and Michael B. Hall[1,*]

1   The University of Queensland, UQ Centre for Clinical Research, QLD 4029, Herston, Australia

## ABSTRACT

High-throughput sequencing datasets frequently exhibit extreme read depth variation, biasing downstream analysis. Normalising coverage to a specific depth cap is important, yet existing tools rely on computationally expensive fetch-based or non-deterministic greedy algorithms. Here, we present a new coordinate-sorted sweep-line algorithm implemented in the open-source software `rasusa` that enforces a strict coverage cap at every genomic position. By utilising seeded random priority assignment, we achieve unbiased, reproducible read selection. The algorithm reduces runtimes by over 1,400-fold compared to legacy fetch-based methods—slashing processing from hours to mere seconds—and operates roughly four times faster than `VariantBam`. Furthermore, it requires only 8 MB of memory for long-read data. This provides a highly efficient, scalable, and reproducible solution for sequencing coverage normalisation.

**Subjects**  Software and Workflows, Bioinformatics, Software Engineering

## STATEMENT OF NEED

Next-generation sequencing datasets frequently suffer from uneven coverage due to PCR duplication, GC bias, or simple abundance variation [1]. For many applications—such as benchmarking variant callers, simulating lower-yield runs, or training deep learning models—it is desirable to normalise data to a uniform depth.

While simple downsampling (subsampling a fixed fraction of reads) is supported by standard tools like `samtools`, this method preserves existing coverage peaks and troughs rather than flattening them. True "capped-depth" downsampling (ensuring no position exceeds coverage $N$) is rarely implemented. To our knowledge, the only other tool offering this capability is `VariantBam` [2]. However, `VariantBam` has seen limited development activity in recent years (last updated circa 2023). Furthermore, users have reported outstanding issues regarding its application to Oxford Nanopore Technologies (ONT) data, highlighting the need for a tool explicitly designed to handle the specific characteristics of modern long-read sequencing.

Naive implementations of capped downsampling often perform repeated random access to the alignment file (BAM/CRAM) to calculate depth at every locus, resulting in severe performance bottlenecks due to I/O latency. There is a need for a maintained, high-performance tool that can enforce coverage caps efficiently, particularly for short read datasets, where read overlaps are small but significant in number. We address this by implementing a streaming sweep-line algorithm within the existing `rasusa` framework [3].

**Submitted:**     26 February 2026

\* Corresponding author. E-mail: michael.hall1@uq.edu.au

Preprint submitted at https://doi.org/10.5281/zenodo.19490078

## IMPLEMENTATION

### Algorithm design

The core innovation in this work is the replacement of fetch-based depth calculations with a coordinate-sorted sweep-line algorithm. The method processes the SAM/BAM/CRAM file sequentially, maintaining an "active set" of reads that cover the current genomic position.

Consistent with the `rasusa` suite's design philosophy, read selection is fundamentally random rather than greedy (e.g., retaining the first $N$ reads). To ensure this randomness is reproducible, we employ a deterministic seeding strategy:

(i) Seeded Priority Assignment: We use a pseudo-random number generator (PRNG) with a user-provided seed (or a random seed if none is provided). As the file is streamed, each read is assigned a random 64-bit integer drawn sequentially from the PRNG. This assigns a priority score that is deterministic given the same seed and input file order.

(ii) Active Set Maintenance: We use a standard binary heap (specifically a max-heap) to track active reads. As the sweep-line advances across the genome, reads ending before the current position are evicted.

(iii) Selection Logic: When a new read is encountered:

- If the active set size is below the target coverage $N$, the read is added.
- If the set is full, the new read's priority is compared against the highest priority currently in the heap. If the new read has a lower (better) priority, we perform a distance check: the swap is only permitted if the start positions of the new read and the evictee are within a configurable `swap_distance` (default 5 bp). This constraint prevents the introduction of coverage gaps in previously processed regions.

This approach guarantees that at any position $i$, the retained reads represent a random sample of size $\min(\text{depth}_i, N)$ with the lowest priority scores.

### Optimised batched-fetching strategy

While the streaming sweep-line algorithm is the primary innovation for coordinate-sorted files, we also overhauled `rasusa`'s existing fetch-based strategy for scenarios where index-based random access (.bai/.crai) is preferred. To overcome the severe I/O bottlenecks of the legacy naive approach, the newly optimised batched-fetching implementation queries and caches reads in large, contiguous genomic windows (default 10 kbp). By shuffling and sampling from these in-memory batches, this method drastically reduces disk-seeking overhead compared to querying locus by locus.

### Processing strategies

#### Long-read

For single-end long reads, we perform a single pass over the file. The decision to keep or discard a read is made instantaneously based on the sweep-line state, and the read is written to the output stream immediately if retained.

#### Paired-end

To maintain read pairing information, we employ a *two-pass* strategy.

**Table 1.** Performance benchmarking results targeting a uniform 50× coverage depth. Mean runtime (*n* = 10 runs) and peak memory were evaluated on a single core.

| Dataset | Implementation | Command | Mean runtime (s) | Peak memory (MB) |
|---|---|---|---|---|
| Illumina | variantBam | variant in.bam -m 50 -b -o out.bam | 118 ± <1 | 116 |
| | rasusa (Naive Fetch) | rasusa aln -s 2109 -c 50 in.bam -o out.bam* | 19369 ± 34 | 22 |
| | rasusa (Batched Fetch) | rasusa aln -s 2109 -c 50 --strategy fetch in.bam -o out.bam | 130 ± 1 | 128 |
| | rasusa (Sweep-line) | rasusa aln -s 2109 -c 50 in.bam -o out.bam | 30 ± 1 | 56 |
| ONT | variantBam | variant in.bam -m 50 -b -o out.bam | 391 ± <1 | 23652 |
| | rasusa (Naive Fetch) | rasusa aln -s 2109 -c 50 in.bam -o out.bam* | 135767† | 97 |
| | rasusa (Batched Fetch) | rasusa aln -s 2109 -c 50 --strategy fetch in.bam -o out.bam | 1227 ± 20 | 314 |
| | rasusa (Sweep-line) | rasusa aln -s 2109 -c 50 in.bam -o out.bam | 95 ± 3 | 8 |

*Command executed using a previous release of rasusa (v2.2.2) that utilised the original naive fetching algorithm.
†Executing 10 replicates exceeded the allocated walltime limit; therefore, the reported runtime represents a single run.

- *Pass 1 (Selection):* The file is scanned to determine which read templates (pairs) should be kept based on the coverage cap. We store the unique identifiers of selected reads in a hash set.
- *Pass 2 (Retrieval):* The file is read a second time. Any read whose name is present in the selection set is written to the output, ensuring that mate pairs are correctly recovered even if they are unmapped, or physically distant in the file.

## VALIDATION

## Performance benchmarking

We benchmarked the new sweep-line implementation against three alternatives: the legacy naive fetch-based implementation in rasusa, the newly optimised batched-fetching implementation (Optimised batched-fetching strategy), and VariantBam [2]. Benchmarks were conducted on the Bunya supercomputer (University of Queensland) [4]. All tests were performed on a standard compute node equipped with dual 48-core AMD EPYC 4th Gen "Genoa" processors (though only a single core was used). The datasets used were high-depth *Salmonella enterica* Illumina paired-end (average depth ~824×; accession SRR26899147) and ONT long read (average depth ~2943×; accession SRR26899102) alignments from the same sample (accession SAMN38321309). The downsampling was targeted to a uniform depth of coverage 50× using a fixed random seed (-s 2109). Run times were measured using hyperfine [5], and peak memory usage was captured via GNU /usr/bin/time -v. The performance improvements delivered by the sweep-line algorithm were striking (Table 1). It dramatically outperformed all other implementations in processing speed, running roughly four times faster than VariantBam on both datasets. Compared to the legacy naive fetch approach, the runtime reductions are remarkable: for the Illumina dataset, processing time fell from nearly 5.4 h (19,369 s) to just 30 s (an over 600-fold improvement). The gains were even more pronounced on the ONT dataset, where the sweep-line method reduced the runtime from 37.7 h (135,767 s) down to just 95 s—an improvement of over 1,400-fold. Furthermore, the memory efficiency of the sweep-line approach on the long-read ONT dataset was impressive, requiring only 8 MB compared to the 23.6 GB consumed by VariantBam. While the legacy naive approach maintained a marginally smaller memory footprint on the Illumina dataset (22 MB versus 56 MB), the sweep-line algorithm provides an overwhelmingly superior balance of speed and memory efficiency.



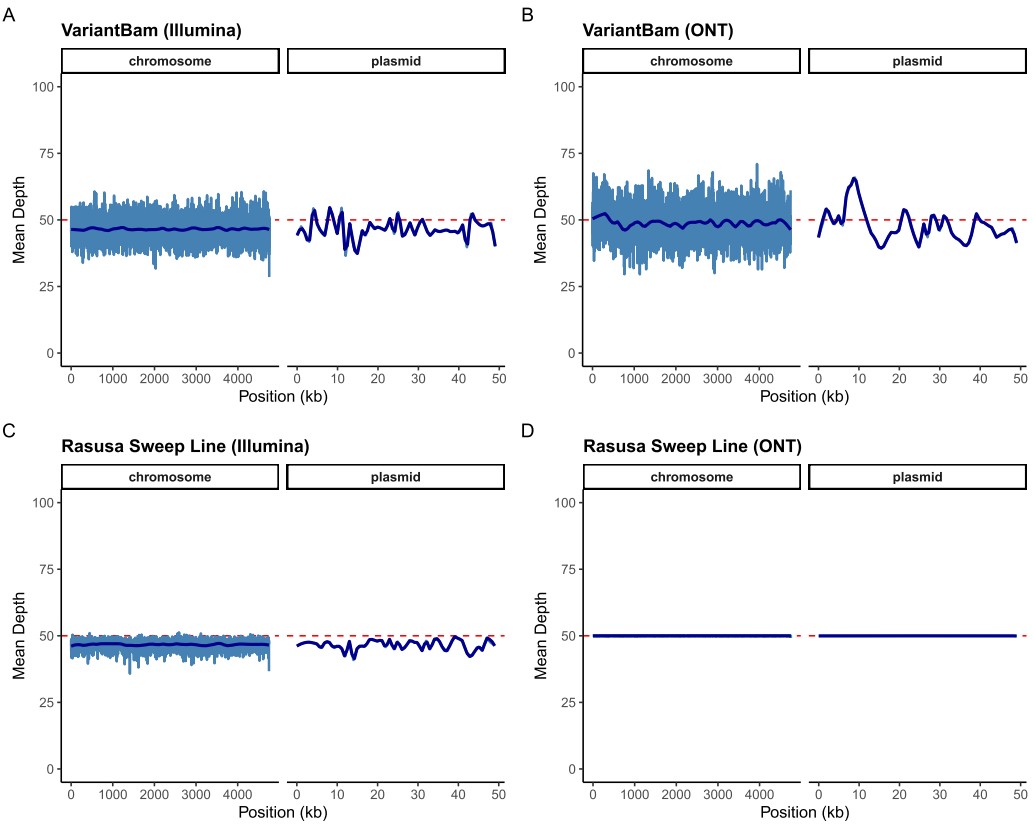

**Figure 1.** Mean coverage depth across the *S. enterica* chromosome and plasmid after downsampling to a 50× target (red dashed line; LOESS smoothed average in dark blue). (A, B) `VariantBam` applied to Illumina paired-end and ONT long read data, respectively. (C, D) `rasusa` (sweep-line algorithm) applied to the same datasets.

## Correctness and bias

To verify correctness, we computed per-position coverage on the downsampled output using `samtools depth` [6]. For long read data, the observed coverage formed a flat line strictly adhering to the requested cap *N* (Figure 1). For paired-end Illumina data, coverage was observed to fluctuate near to, but rarely exceed, the target cap. This fluctuation is an expected consequence of three factors: the `swap_distance` constraint, which limits the search space for optimal read replacement to preserve local continuity; the presence of unmapped mates, which results in singleton alignments that do not contribute to coverage in their expected mate's region; and natural variation in template lengths, meaning the physical distance between the first and last segment is not identical across pairs, which makes it difficult to perfectly optimise the coverage cap simultaneously at both mate loci. The `VariantBam` depth fluctuated considerably more, particularly on the ONT data (Figure 1B).

## AVAILABILITY OF SOURCE CODE AND REQUIREMENTS

- Project name: Rasusa
- Project homepage: https://github.com/mbhall88/rasusa
- Operating system: Platform independent



- Programming language: Rust
- Other requirements: None (Rust toolchain required only if building from source)
- License: MIT license.

The algorithm described here is implemented in version 3.0.0.

## DATA AVAILABILITY

The raw sequencing data supporting the results of the article are available in the European Nucleotide Archive (ENA) under accession numbers SRR26899147 (Illumina paired-end) and SRR26899102 (ONT). The *S. enterica* ATCC 10708 reference genome used for alignment was RefSeq GCF_045287905.1 [7].

To generate the high-depth BAM files used for the benchmarking, Illumina reads were aligned against the reference genome using Bowtie2 [8] with the default settings. The ONT reads were aligned using minimap2 [9] using options `-aL -cs -MD -t 8 -x map-ont --secondary=no`. All resulting alignments were coordinate-sorted and converted to BAM format using `samtools sort` [6].

## LIST OF ABBREVIATIONS

ONT, Oxford Nanopore Technologies; PRNG, pseudo-random number generator.

## DECLARATIONS

### Ethical approval

The authors declare that ethical approval was not required for this type of research.

### Competing interests

The authors declare that they have no competing interests.

### Author's contributions

ADCF: Conceptualisation, Methodology, Software, Validation, Formal analysis, Investigation, Data Curation, Writing - Original Draft, Writing - Review & Editing, Visualisation. LWR: Conceptualisation, Writing - Review & Editing, Supervision, Funding acquisition. MBH: Conceptualisation, Methodology, Software, Investigation, Data Curation, Writing - Original Draft, Writing - Review & Editing, Supervision.

### Funding

This research was supported by a National Health and Medical Research Council (NHMRC) Investigator grant [Grant ID: 2026911]. ADCF was supported by a scholarship from the Indonesia Endowment Fund for Education Agency (LPDP), Ministry of Finance, Republic of Indonesia.

### Acknowledgements

This work was supported by resources provided by The University of Queensland Research Computing Centre's Bunya supercomputer [4], with funding from The University of Queensland, Brisbane, Australia.

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
