## [Editor Report]

Editor’s AssessmentThe manuscript is ready for formal accept.Editor’s AssessmentThe manuscript is ready for formal accept.

---

## [Reviewer Report]

Reviewer name and names of any other individual's who aided in reviewerDaniel ParkDo you understand and agree to our policy of having open and named reviews, and having your review included with the published manuscript. (If no, please inform the editor that you cannot review this manuscript.)YesIs the language of sufficient quality?YesPlease add additional comments on language quality to clarify if neededIs there a clear statement of need explaining what problems the software is designed to solve and who the target audience is? YesAdditional CommentsIt is true that there really isn't other good, performant, well maintained software out there that addresses this specific need.Is the source code available, and has an appropriate Open Source Initiative license <a href="https://opensource.org/licenses" target="_blank">(https://opensource.org/licenses)</a> been assigned to the code?YesAdditional CommentsAs Open Source Software are there guidelines on how to contribute, report issues or seek support on the code?YesAdditional CommentsIs the code executable?YesAdditional CommentsI have extensively tested it myself.Is installation/deployment sufficiently outlined in the paper and documentation, and does it proceed as outlined?YesAdditional CommentsI have only really tested the bioconda installation path myself but that path works flawlessly on both Intel and ARM architectures.Is the documentation provided clear and user friendly?YesAdditional CommentsIs there enough clear information in the documentation to install, run and test this tool, including information on where to seek help if required?YesAdditional CommentsIs there a clearly-stated list of dependencies, and is the core functionality of the software documented to a satisfactory level?YesAdditional CommentsYep, dependencies are all named in the Cargo.toml, usage and invocation documented in README.mdHave any claims of performance been sufficiently tested and compared to other commonly-used packages? YesAdditional CommentsIs test data available, either included with the submission or openly available via cited third party sources (e.g. accession numbers, data DOIs)?YesAdditional CommentsAccessions and methods to reproduce the benchmarking analyses in this manuscript are provided at the end.Are there (ideally real world) examples demonstrating use of the software? YesAdditional CommentsIs automated testing used or are there manual steps described so that the functionality of the software can be verified?YesAdditional CommentsA comprehensive test suite of rust tests exists as Github Actions workflows and CodeCov (https://app.codecov.io/github/mbhall88/rasusa) reports 70% coverage on the code base.Any Additional Overall Comments to the AuthorApologies for the delayed review. I wanted the opportunity to do some testing and benchmarking myself. I happened to have handy a pretty extreme dataset that exemplifies the need for a tool like this, and was extremely happy with the results (and speed!). My analysis is consistent with the findings of the authors and is described in further detail here: https://github.com/broadinstitute/viral-ngs/pull/1059#issuecomment-4180420502 The rasusa v3 tool effectively downsampled in the areas of highest oversequencing while generally maintaining coverage for other parts of the genome. I was also curious a bit about the "collateral damage" effects of reduced genome coverage nearby the capped peaks--the final plots on my analysis show this to be essentially contained within the library fragment length distance from the downsampled regions, which is reassuring, and we experienced very minimal effects on the total number of bases with callable variants (first plots). Given the insane speed of this tool with minimal downside, it's easy to choose to simply always run this in one's pipelines. I am also appreciative of the author's polling of the ubioinfo community during development and choosing to include the second-pass-mate-pair-rescue step based on community feedback prior to releasing v3 of this software.RecommendationAccept

---

## [Reviewer Report]

Reviewer name and names of any other individual's who aided in reviewerWouter De CosterDo you understand and agree to our policy of having open and named reviews, and having your review included with the published manuscript. (If no, please inform the editor that you cannot review this manuscript.)YesIs the language of sufficient quality?YesPlease add additional comments on language quality to clarify if neededIs there a clear statement of need explaining what problems the software is designed to solve and who the target audience is? YesAdditional CommentsIs the source code available, and has an appropriate Open Source Initiative license <a href="https://opensource.org/licenses" target="_blank">(https://opensource.org/licenses)</a> been assigned to the code?YesAdditional CommentsAs Open Source Software are there guidelines on how to contribute, report issues or seek support on the code?YesAdditional CommentsIs the code executable?YesAdditional CommentsIs installation/deployment sufficiently outlined in the paper and documentation, and does it proceed as outlined?YesAdditional CommentsIs the documentation provided clear and user friendly?YesAdditional CommentsIs there enough clear information in the documentation to install, run and test this tool, including information on where to seek help if required?YesAdditional CommentsIs there a clearly-stated list of dependencies, and is the core functionality of the software documented to a satisfactory level?YesAdditional CommentsHave any claims of performance been sufficiently tested and compared to other commonly-used packages? YesAdditional CommentsIs test data available, either included with the submission or openly available via cited third party sources (e.g. accession numbers, data DOIs)?YesAdditional CommentsAre there (ideally real world) examples demonstrating use of the software? YesAdditional CommentsIs automated testing used or are there manual steps described so that the functionality of the software can be verified?YesAdditional CommentsAny Additional Overall Comments to the AuthorRecommendationAccept